# Role of Filamin A in Growth and Migration of Breast Cancer—Review

Patryk Zawadka [1,†], Wioletta Zielińska [1,†], Maciej Gagat [1,2,*] and Magdalena Izdebska [1]

1   Department of Histology and Embryology, Faculty of Medicine, Collegium Medicum in Bydgoszcz, Nicolaus Copernicus University in Toruń, 85-092 Bydgoszcz, Poland; patryk.zawadka@cm.umk.pl (P.Z.); w.zielinska@cm.umk.pl (W.Z.); mizdebska@cm.umk.pl (M.I.)
2   Faculty of Medicine, Collegium Medicum, Mazovian Academy in Płock, 09-402 Płock, Poland
*   Correspondence: mgagat@cm.umk.pl
†   These authors contributed equally to this work.

**Abstract:** Despite ongoing research in the field of breast cancer, the morbidity rates indicate that the disease remains a significant challenge. While patients with primary tumors have relatively high survival rates, these chances significantly decrease once metastasis begins. Thus, exploring alternative approaches, such as targeting proteins overexpressed in malignancies, remains significant. Filamin A (FLNa), an actin-binding protein (ABP), is involved in various cellular processes, including cell migration, adhesion, proliferation, and DNA repair. Overexpression of the protein was confirmed in samples from patients with numerous oncological diseases such as prostate, lung, gastric, colorectal, and pancreatic cancer, as well as breast cancer. Although most researchers concur on its role in promoting breast cancer progression and aggressiveness, discrepancies exist among studies. Moreover, the precise mechanisms through which FLNa affects cell migration, invasion, and even cancer progression remain unclear, highlighting the need for further research. To evaluate FLNa's potential as a therapeutic target, we have summarized its roles in breast cancer.

**Keywords:** filamin A; breast cancer; metastasis; cell migration; cell invasion

## 1. Introduction

Breast cancer is a challenge for the healthcare system, as it remains the leading cause of cancer-related deaths in young women. Despite the constant advancements in research, diagnosis, and increasing public awareness, morbidity suggests that little or no progress in preventing the disease has been made [1,2]. The survival rate for women with primary tumors is high, but the chances rapidly drop as soon as metastasis begins. The median survival time after the diagnosis of metastasis is around 24 months. It is supposed that only 5–10% of cases are genetically related, while the majority—90–95%—are connected with the lifestyle or environment [3]. The risk factors for breast cancer include age, alcohol consumption, hormonal therapy, and history of cancers in the family [3].

Quick diagnosis and choice of appropriate therapy are crucial in treating patients with breast cancer. In patients with primary tumors, the main therapeutic strategy is surgical excision. However, once the process of metastasis begins, new therapeutic challenges arise. Breast cancer represents a spectrum of oncological disorders, requiring treatment options to be specifically tailored to each unique case [1,2,4]. Researching new treatment approaches is especially essential for triple-negative breast cancer, which is resistant to the most conventional therapies. These approaches may encompass novel medications or targeted therapies targeting proteins that exhibit abnormal expression levels in cancer cells [2,4].

One such protein may be filamin A (FLNa), which belongs to actin-binding proteins (ABPs) that interact with the actin cytoskeleton and allow its reorganization. In turn, the rearrangement of the actin cytoskeleton is a fundamental process for controlling

both cell morphology and movement. FLNa plays a pivotal role in forming networks of actin filaments within the cortical cytoplasm and anchors membrane proteins to the actin cytoskeleton. It also interacts with integrins, transmembrane receptor complexes, and secondary messengers, which regulate cell migration and adhesion [5–7]. The exact role of FLNa in oncological diseases remains controversial. Initially, FLNa was identified as a cancer-promoting protein implicated in invasion and metastasis [7,8]. In turn, other research has shown that FLNa impedes tumor formation or progression, casting uncertainty on the exact role of FLNa in cancer development [8,9].

FLNa's role has been identified in a wide range of cancer types, including prostate, pancreatic, lung, colorectal, gastric, and breast cancers when compared to healthy tissues [10]. Another notable aspect of the protein's function is its cellular localization, which may also affect cancer progression. Research indicates that elevated cytoplasmic levels of the protein are associated with increased cell invasiveness, migration, and adhesion. In turn, its presence in the nucleus impacts gene expression and diminishes the capacity for cell migration [11]. There is evidence to suggest that in the case of breast cancer, cyclin D1 interacts with FLNa, which promotes the migration and invasive potential of cancer cells [12]. On the other hand, FLNa has also been found to regulate focal adhesion disassembly and inhibit breast cancer cell migration and invasion [9]. To systematize the numerous reports regarding the involvement of FLNa in breast cancer, our article comprehensively assesses and reconciles existing findings on FLNa's involvement in breast cancer development and progression.

## 2. Breast Cancer

Because of its heterogeneity, breast cancer poses a huge challenge in treatment. Based on its molecular characteristics, it can be divided into three groups: hormone-positive (either with the expression of estrogen receptors ($ER^+$) or progesterone receptors ($PR^+$)), with the expression of human epidermal receptor 2 ($HER2^+$), and triple-negative breast cancer (TNBC), which does not show expression of hormone receptors and HER2 ($ER^-$, $PR^-$, $HER2^-$) [2,4].

$ER^+$ is the most common breast cancer type and makes up for 60–70% of cases in premenopausal women in developed countries. It can be divided into two subcategories—luminal A and B. The treatment regimen for $ER^+$ breast cancers usually includes hormone therapy with estrogen blockers/aromatase inhibitors as the main therapeutic agents (2). $HER2^+$ breast cancer accounts for 20% of cases with a therapeutic strategy based mostly on chemotherapy and anti-HER2 monoclonal antibodies. Even though it improved overall survival, the problem of the disease reoccurrence still stands [1,2]. TNBC is the rarest of the breast cancer types, as it is diagnosed in 10–20% of patients. It can be divided into six subcategories: basal-like 1 (BL-1), basal-like 2 (BL-2), immunomodulatory (IM), mesenchymal (M), mesenchymal stem cell-like (MSL), and luminal androgen receptor (LAR). As TNBC does not express any of the receptors the previous groups do, the therapeutic approach remains the most challenging. As conventional hormonal or monoclonal antibody therapies do not work, the researchers are forced to look for new drugs/targetable pathways [4,13].

Breast cancer cells pose a significant threat due to their remarkable ability to convert from an epithelial to a more mobile mesenchymal phenotype during a complex process known as epithelial–mesenchymal transition (EMT). This transition is a key driver of breast cancer aggressiveness, enabling them to infiltrate the bloodstream, access distant regions of the body, and initiate invasion [14,15]. In epithelial cells, lamellipodia and filopodia are the most common types of actin projections, whereas invadopodia are predominantly found in mesenchymal cells [14]. Although it is established that early metastasis in patients with TNBC contributes to the grim prognosis, the full mechanism underlying its spontaneous onset remains incompletely understood [14,15].

### 3. Filamin A and Its Functions

The family of filamins consists of three highly homologous proteins: FLNa, filamin B (FLNb), and filamin C (FLNc). They are large cytoplasmic ABPs that crosslink F-actin. Contrary to FLNa and FLNb, which are expressed in many tissues, FLNc is mostly limited to skeletal muscle and myocardium. Apart from their crosslinking abilities, FLNs serve as a scaffold for signaling proteins, i.e., tyrosine kinases, phosphatases, or GTPases, as well as for adhesive receptors, like integrins [7,16,17].

FLNa plays several vital roles inside the cell (Figure 1). As a cytoplasmic protein, FLNa significantly influences the branching of F-actin and plays a crucial role in anchoring transmembrane proteins to the actin cytoskeleton, which also impacts cell adhesion [18]. The interaction between FLNa and actin induces swift modifications in actin filaments, consequently reshaping the cytoskeleton into orthogonal networks [7,17]. These FLNa-actin networks exhibit dynamic and reversible organizational characteristics, safeguarding the cell against diverse shear stresses. Nonetheless, the precise function of FLNa in cellular motility and adhesion remains a subject of debate and could hinge on factors such as protein expression levels or its interacting partners. In this regard, talin appears to be a particularly intriguing protein, as filamins, including FLNa, and talin share the same binding site on integrins. These two proteins vie for binding to integrin tails, giving rise to integrin–FLNa interactions that can influence talin-dependent integrin activation [19]. This phenomenon influences talin-dependent processes like integrin activation. The binding of talin to integrin tails is a crucial step in integrin activation and the forming of integrin–cytoskeletal links [20]. Furthermore, integrin threonine phosphorylation causes inhibition of FLNa binding to integrins without affecting talin. This process could be a way to control the binding of FLNa and the competition between FLNa and talin [20]. Mutations mimicking phosphothreonine hinder the binding of FLNa but not talin, suggesting that kinases may regulate this competition, offering an additional avenue to modulate integrin functions [19]. The dynamics of interactions involving FLNa, talin, and integrin are intricate, yet it is evident that they influence cell adhesion and migration [19]. Calderwood et al. demonstrated that enhanced FLNa binding to the beta-integrin cytoplasmic domain inhibited cell migration and the formation of membrane protrusions but did not affect fibronectin matrix assembly or focal adhesion formation [21]. Consequently, robust FLNa binding curtails integrin-dependent cell migration by impeding transient membrane protrusion and cell polarization [21]. Furthermore, loss-of-function mutations in FLNa are implicated in impaired neural cell migration in response to microenvironmental cues, along with the development of vascular system defects [22–26].

FLNa plays a role in cell proliferation. In mammalian cells, the initiation of mitosis involves a heterodimer of cyclin B1 and the cyclin-dependent kinase 1 (Cdk1). Cukier et al. demonstrated that cyclin B1 forms a complex with FLNa, as the proteins co-immunoprecipitate and co-localize in mitotic human cells [27]. Moreover, mitotic phosphorylation of FLNa is involved in the actin rearrangement during cell division, formation of the actin contractile ring, and separation of daughter cells, indicating that FLNa is crucial for successful cell division [27,28].

FLNa also takes part in cell invasion. FLNa can bind both F- and G-actin, which allows it to form cross-linked networks of bundles from actin filaments [29]. Due to its structure, it can create flexible connections between actin filaments, which are necessary to form invasive structures like filopodia and lamellipodia [29]. Additionally, FLNa activates various kinases and is subject to regulation by kinases itself [30]. FLNa is one of the most extensively studied substrates of p21-activated kinase-1 (PAK1), a serine/threonine-protein kinase with a profound impact on diverse cellular processes, including directional motility, invasion, metastasis, growth, cell cycle progression, and angiogenesis [31,32]. PAK1 is frequently overexpressed and hyperstimulated in various human cancers [32]. The interaction between FLNa and PAK1 amplifies the kinase activity of PAK1. This interaction leads to the phosphorylation of FLNa at serine (Ser) 2152, inducing PAK1-dependent membrane ruffling [31]. Phosphorylation of FLNa by binding with PAK1 causes it to

activate PAK1, which allows for PAK1 downstream actin-modulating signaling in a positive feedback loop [33]. FLNa interacts with sphingosine kinase 1, which phosphorylates sphingosine, which activates PAK1 [33].

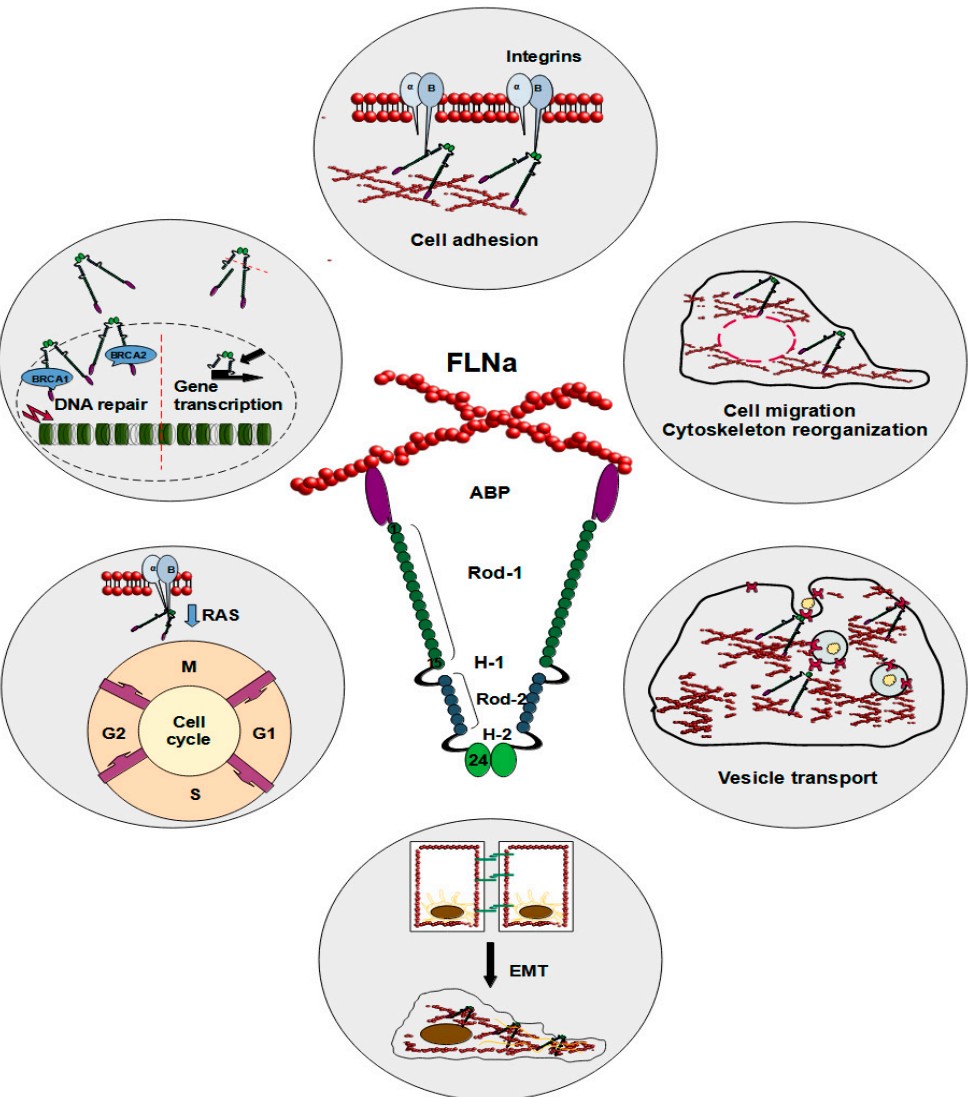

**Figure 1.** Schematic illustration of FLNa roles in a cell. The reported functions of FLNa include cell adhesion, cell migration, cytoskeleton reorganization, vesicle transport, epithelial–mesenchymal transition, cell cycle progression, DNA repair, and gene transcription.

Ser 2152 phosphorylation inhibits the cleaving of FLNa at its hinge region and is speculated as being the reason for its localization in the cytoplasm [34]. Cytoplasmatic localization allows FLNa to regulate numerous signaling proteins, for example, mitogen-activated protein kinases (MAPKs) [34].

Apart from breast cancer, FLNa is also present in various types of malignancies. Depending on the type of cancer, its expression may be down- or upregulated [7]. Cancers with high FLNa expression include, but are not limited to, breast cancer, which is the main focus of this review and will be investigated in detail subsequently, melanoma, and glioblastoma [7]. On the other hand, the malignancies with low FLNa expression are colorectal, bladder, and prostate cancers [7]. The abundance of cancers with different FLNa expressions makes it a perfect target for studies and, later on, targeted therapy.

The exact mechanisms and pathways of FLNa influencing cell proliferation, migration, and invasion remain unknown. The available reports suggest some possible explanations

for the role of FLNa in these processes. One of the first explanations was that FLNa can influence epithelial–mesenchymal transition (EMT), during which cells undergo a phenotypic shift from a non-invasive epithelial state to a mesenchymal cell state, enabling cell movement [35,36]. Studies performed on colorectal cancer and adenocarcinoma showed that increased expression of FLNa enhances EMT [35,36]. In adenocarcinoma, suppression of FLNa hinders snail-induced cell adhesion, diminishes the size of focal adhesion sites, prompts the relocation of talin from the cytoplasm to the membrane area, and enhances cell migratory properties [36].

This chapter outlines the most significant proteins associated with FLNa that hold importance in the context of cancer. It is important to acknowledge that the coverage here is not exhaustive. Reports indicate an abundance of FLNa-interacting partners, including adapter proteins, small GTPases, transmembrane receptors, and membrane channels [37,38].

## 4. Role of Filamin A in Breast Cancer

### 4.1. The Expression of Filamin A in Breast Cancer

Multiple studies have confirmed the expression of FLNa protein in breast cancer specimens. Joshua et al. analyzed 100 cases of breast cancer samples, finding that FLNa was present in 69% of the most invasive cases [39]. They also noted the expression of the protein in normal tissue and non-malignant carcinoma. Tian et al. conducted a similar investigation on 96 samples of breast cancer tissue, confirming FLNa's presence in 63.5% of the cases [40]. Simultaneously, in distant normal breast tissues and benign tumor samples, FLNa was almost undetectable. The percentage of FLNa-positive samples increased with the cancer's progression—showing up in 36% of stage I, 66.7% of stage II, and 84.6% of stage III cases [40]. Studies conducted by Guo et al. showed that 52.6% of the samples displayed the protein expression. In contrast, FLNa was hardly noticeable in normal breast tissues and benign tumors, with a mere 5.9% of normal breast tissue samples showing positive staining results [41]. In turn, Jiang et al. noted that only 39.9% of the breast cancer samples exhibited FLNa protein expression [42]. The reason may be that research focused exclusively on early-stage (I/II) breast cancer patients. Additionally, there were some variations across the studies regarding the criteria used to classify staining results as positive or negative.

The researchers observed variations in the levels of FLNa expression between cancer and normal tissues and changes in its distribution pattern. Joshua et al. observed exclusively cytoplasmic FLNa expression across all the breast cancer samples. In contrast, in adjacent normal breast tissue and benign fibroadenoma tissues, FLNa expression was primarily membranous with a slight cytoplasmic positive signal [39]. Similarly, Tian et al. reported that in cancer tissues, FLNa was predominantly located in the cytoplasm, especially at the cell edges, and was also present in basal cells and intercellular matrix [40]. This aligns with the observations by Guo et al., who confirmed that the FLNa protein was primarily found in the cytoplasm of myoepithelial and basal cells or between cells in breast cancer tissue [41]. Cytoplasmic localization of FLNa in breast cancer tissue samples was also noted in studies by Jiang et al. [42]. Moreover, in the MDA-MB-231 invasive breast cancer cell model, a positive signal for FLNa was accumulated within the cytoplasm and at the cell membrane [12].

While existing reports concur regarding the percentage of FLNa-positive breast cancer samples and the predominantly cytoplasmic expression observed in cancer cells, the association between FLNa protein levels and clinicopathological parameters raises more questions. Joshua et al. did not confirm any significant relationships between FLNa expression and histological grade, age, parity, oral contraceptive use, smokeless tobacco use, TNM stage, clinical stage, clinical prognostic staging, as well as estrogen and estrogen receptors, Her2/neu, and ki-67 status. This could point to the potential of FLNa as an independent biomarker in breast carcinoma [39]. In turn, Tian et al. noted that the FLNa protein level correlated with the TNM stage, presence of lymph node metastases, occurrence of vascular or neural invasion, and menstrual status. On the other

hand, FLNa expression showed no significant relationship with other clinicopathological characteristics such as patient age, tumor dimensions, location, histological classification, the estrogen and progesterone receptors, and the Her2/neu status [40]. In a report by Guo et al. the level of FLNa protein expression correlated negatively with tumor size. Conversely, a higher incidence of positive FLNa staining was observed alongside increased expression of progesterone receptors. There was no discernible link between the expression levels of FLNa protein and factors such as patient age, lymph node metastasis, tumor differentiation, estrogen receptor status, or Her2/neu protein expression [41]. There is also a lack of consensus regarding the impact of FLNa expression on patients' survival. In one report, the absence of FLNa expression correlated with longer patient survival without distant metastasis, whereas another report finds that high levels of FLNa expression are associated with extended patient survival [41].

There are also several reports on how the manipulation of interacting partners impacts FLNa expression. Ljepoja et al. carried out a genomic knockout of miR-200c in MCF-7 breast cancer cells [43]. The team noted that over 50% of all diversely expressed proteins in cells were linked to cells' migration and that FLNa was one of the most eminent. Researchers also checked the biological effect of miR-200c on MDA-MB-231 TRIPZ-200c and MDA-MB-231 TRIPZ-Ctrl cells, which were transduced with a TET-off construct containing miR-200c or a scrambled control, respectively. They observed that miR-200c downregulated FLNa, which matched their previous study on MCF-7 cells with miR-200c knocked out (MCF-7-200c-KO) [43]. Both the mRNA and protein levels of FLNa decreased, by 30% and 40%, respectively, compared to TRIPZ-Ctrl cells stimulated with doxycycline. Knockout of miR-200c in MCF-7 showed an increase in cellular expression of FLNa, while induction of miR-200c in MDA-MB-231 TRIPZ-200c resulted in FLNa downregulation [43]. Among the proteins that functionally correlate with FLNa is FLNa-interacting protein 1-like (FILIP1L), an inhibitor of cell migration and invasion of ovarian cancer, found in various cell lines, including breast cancer cells as well as normal human mammary epithelial cells (HMECs) [44]. FILIP1L was shown to inhibit migration via FLNa degradation in neocortical cells [45]. Kwon et al. noted that the level of FILIP1L in MCF-7 cells was as high as in normal HMECs. It was also stated that the protein expression is linked with its mRNA expression and that it can be downregulated in various cancer cell lines compared to normal or non-invasive cancer cell lines [44].

### 4.2. Role of Filamin A in Cell Migration and Invasiveness of Breast Cancer Cells

There are numerous reports on the role of FLNa in cells' migration and invasiveness. Most researchers concluded that overexpression of FLNa contributes to increased motility and invasive properties of breast cancer cells [8,42,46,47]. Shao et al. noted the duality of FLNa's role in the metastasis of breast cancer [8]. On the one hand, it serves as a regulator of focal adhesion disassembly and a suppressor for cell migration and invasion via interaction with caveolin-1 [8]. Furthermore, its deficiency in MDA-MB-231 cells lowers their migration and invasion ability via the interaction with cyclin D1 [12]. On the other hand, breast cancer cells' motility was increased by upregulating phosphorylated PAK1 through FLNa [8]. The researchers stated that FLNa is overexpressed in metastatic breast cancer and its cytoplasmic localization correlates with metastasis to lymph nodes, advanced stage, invasion of blood vessels or nerves, state of menstruation, and other risks. Moreover, lower levels of FLNa correlated with improved distant metastasis-free survival [8]. Ji et al. stated that FLNa is overexpressed in breast cancer and colocalizes in the cytoplasm with 14-3-3σ—a multifunctional protein present in most eukaryotic cells and is involved in many cell functions like regulating cytoskeleton, transducing signal, membrane signaling, apoptosis, adhesion, proliferation, and differentiation [47].

Multiple studies focused on the effect of FLNa downregulation in breast cancer cells, with various results. Zhou et al. knocked out FLNa in MDA-MB-231 cells in two target sites signed as FLNA/KO-1 and FLNA/KO-2. It decreased cell proliferation after 72 h by 76.35% and 75.61%, respectively, compared to the MDA-MB-231 negative control [46]. Cell

migration rate in wound healing was also affected and dropped to 43.95% and 43.84%, respectively, compared to the control group. What is more, the ability of breast cancer cells for migration and invasion decreased drastically to 91.17% and 87.06% for the FLNA/KO-1 group and 76.43% and 75.48% for the FLNA/KO-2 group, respectively, compared to the MDA-MB-231 negative control. The researchers concluded that FLNa knockout is not fatal in breast cancer cells, but it severely represses cell proliferation, invasion, and migration capabilities [46]. Ji et al. confirmed that the transfection of MDA-MB-231 cells with FLNa siRNA significantly decreased the expression of both the protein and mRNA [47]. The researchers also checked 14-3-3σ expression after FLNa downregulation and discovered that it led to its rise in both protein and mRNA levels. The study also investigated the effect of silencing FLNa on both the migration and invasion of breast cancer cells and showed that it hinders both processes [47]. Jiang et al. also noted that MDA-MB-231 and MDA-MB-436 cells with FLNa deficiency are not as mobile as the cells with FLNa overexpression, with significant changes between the control and knocked-down cell lines [42]. Downregulation of FLNa significantly hindered the cancer cells' ability to migrate and invade other tissues [42]. Furthermore, Kwon et al. studied the impact of FLNa-interacting protein FILIP1L on the migration of MCF-7 cells [44]. They confirmed that DNA methylation at the FILIP1L promoter resulted in decreased protein expression and limited cell invasiveness [44].

The FLNa expression and its impact on breast cancer cell migration and invasiveness were also tested in animal models. Zhou et al. xenografted mice with wild-type MDA-MB-231, FLNA/NC (negative control), FLNA/KO-1, and FLNA/KO-2 cells [46]. Compared to mice xenografted with wild-type and negative control cells, the tumor volume and growth rate were significantly limited in knockout groups and dropped by 61.72% and 68.30% after 28 days [46]. No metastasis occurred in FLNA/KO groups, either local or distant. Jiang et al. also compared the effect of FLNa expression on tumor growth in a murine model [42]. The study showed that mice injected with FLNa-deficient breast cancer cells showed less frequent metastasis and limited weight loss compared to mice injected with non-transfected control breast cancer cells [42].

Zhou et al. stated that knocking out FLNa had no significant effect on EMT, but it may affect the metastasis process through a different pathway [46]. They suggested that FLNa stimulates matrix metalloproteinases (MMPs)—a group of enzymes that degrade extracellular matrix (ECM) components [46]. The researchers confirmed that FLNa knockout caused a significant reduction in MMP-1 expression but did not influence MMP-2 or MMP-9. Although this mechanism is not yet known, based on the presence of FLNa in both the nucleus and cytoplasm, the research team hypothesized that a fragment of FLNa cleaved by calpain called FLNA-C acts as a transcription factor and direct or indirect promotor of MMP-1. An alternative explanation was that the protein interacts with integrin beta-1 (ITGB1), which, in turn, can bind to various elements of ECM, influencing cell adhesion, invasion, and ECM degradation [46].

Membrane ruffle formation and cell migration require signal-dependent phosphorylation of FLNa on Serine-2152 via PAK1 and ribosomal S6 kinase (RSK). Ravid et al. demonstrated that the expression of caveolin-1 leads to an increase in FLNa Ser 2152 phosphorylation, both under basal conditions and when stimulated by IGF-I, which in turn enhances the migration of MCF-7 cells in response to IGF-I [48]. They reported a significant enhancement in cell migration capabilities with a ten-fold increase in the ability of MCF-7/Cav1 cells to migrate across collagen IV-coated filters. In contrast, the wild-type MCF-7 cells and the control cells containing an empty vector (MCF-7/pJB20) exhibited less than a two-fold increase in cell migration. Similarly, under conditions using vitronectin-coated filters, the introduction of IGF-I led to a nine-fold increase in migration for MCF-7/Cav1 cells, while the MCF-7 and MCF-7/pJB20 cells showed an approximately two-fold increase. Ravid et al. concluded that caveolin-1 expression significantly enhances the migration of MCF-7 cells toward IGF-I through the Akt-mediated phosphorylation of FLNa at Ser-2152. Moreover, the researchers noted that when using a fibroblast-conditioned medium as a

chemoattractant, the migration rates of MCF-7/Cav1 and MCF-7 cells were similar, suggesting a crucial role for caveolin-1 in stimulating cancer cell migration, potentially through its ability to enhance FLNa phosphorylation [48].

### 4.3. Role of Filamin A in DNA Repair and Therapy Outcome in Breast Cancer

Numerous indications suggest FLNA's involvement in cell proliferation and DNA repair. Among interacting partners of FLNa are the BRCA1 and BRCA2 proteins that participate in DNA repair. It has been confirmed that FLNa-lacking cells show lengthened checkpoint activation that leads to the accumulation of cells in G2/M after ionizing radiation [49]. Velkova et al. identified the region of BRCA1 that is mainly responsible for interacting with FLNa—Motif2 [50]. Analyzing data from the Breast Cancer Information Core (BIC), the researchers selected one of the most frequent missense mutations within Motif2, the Y179C variant. Introducing this mutation diminished the interaction within the full-length context compared to the wild-type BRCA1. This analysis suggested that the absence of FLNa could impede the formation of BRCA1 foci following DNA damage. Drawing on their findings and reviewing existing literature, the authors proposed that the Y179C mutation may act as a hypomorphic mutation, exerting moderate effects on the predisposition to breast cancer [50].

The involvement of FLNa in DNA repair is also related to drug resistance. Doxorubicin and docetaxel, used in breast cancer therapy, have shown promising results when combined in clinical trials [51]. Doxorubicin intercalates with DNA, which induces apoptosis via DNA synthesis inhibition, which is very effective against solid tumors [52]. On the other hand, docetaxel is a microtubule inhibitor [53]. Microtubule inhibition causes immunogenic cell death through endoplasmic reticulum stress, which leads to calreticulin translocation to the plasma membrane, which induces phagocytosis [54]. Zhao et al. noted that overexpression of FLNa increases resistance to docetaxel but not doxorubicin [55]. FLNa knockdown in MDA-MB-231, HCC38, Htb126, and HCC1937 cell lines increased their sensitivity to docetaxel. Zhao et al. checked if FLNa influences chemosensitivity through the activation of ERK1/2 in the murine model [55]. They observed that docetaxel therapy was more effective in mice with FLNa deficiency compared to the control group. They also confirmed that the suppression of FLNa led to decreased ERK phosphorylation, while FLNa reintroduction resulted in the activation of ERK. Thus, they concluded that FLNa influences chemoresistance through the MAPK/ERK pathway [55]. Mouron et al. showed that cyclin-dependent kinase (CDK4) and FLNa overexpression correlated with higher sensitivity to paclitaxel [56]. MDA-MB-231 cells overexpressing CDK4 exhibited higher FLNa levels compared to wild-type cells. In turn, FLNa overexpression in MDA-MB-231 cells did not increase CDK4 levels. Noteworthily, MDA-MB-231 with CDK4 or FLNa overexpression showed similar sensitization to paclitaxel. Mouron et al. further observed that FLNa downregulation in MDA-MB-231 cells overexpressing CDK4 restored their sensitivity to paclitaxel. Thus, the researchers conclude that FLNa plays a crucial role in enhancing paclitaxel sensitivity [56].

### 4.4. Role of Filamin A in the Regulation of Gene Expression in Breast Cancer

The cytoplasmic localization and functions of filamin A (FLNa) are well-documented, yet it is increasingly evident that FLNa also resides within the cell nucleus. In metastatic cancers, particularly breast and prostate cancers, FLNa is frequently overexpressed in its cytoplasmic form. Notably, in metastatic prostate cancer, the specific phosphorylation of cytoplasmic FLNa at S2152 inhibits its cleavage into a 90 kDa fragment, a form that is commonly found in the nucleus of less aggressive or benign tumors. It suggests that the expression level, form, and localization of FLNa are all critical factors in cancer development and metastasis.

Research further indicates that both the 90 kDa cleaved fragment and the full-length FLNa can penetrate the cell nucleus, influencing the transcription of various genes. For example, a study by Guo et al. uncovered a positive correlation between FLNa levels and the tumor suppressor protein BRCA1, hinting at FLNa's regulatory role in tumor suppression gene expression [41].

Ravid et al. investigated FLNa levels and phosphorylation at Ser-2152 in IFN-I-treated MCF-7 cells and caveolin-1 expressing transfected MCF-7 cells (MCF-7/Cav1) [48]. They discovered that FLNa was significantly upregulated in MCF-7/Cav1 cells. Moreover, FLNa levels were found to be 2.5 times higher in MCF-7/Cav1 cells compared to the wild-type MCF-7 cells, as confirmed by Western blot analysis. Interestingly, this upregulation of FLNa mRNA and protein was not attributed to the transcriptional activity of the FLNa gene, as caveolin-1 did not induce an FLNa promoter-luciferase reporter [48]. Furthermore, they observed an enhanced basal level of phosphorylated FLNa at Ser-2152 in MCF-7/Cav1 cells, with IGF-I treatment further amplifying FLNa phosphorylation in both cell types, especially in MCF-7/Cav1 cells which already exhibited high levels. The ratio of phosphorylated FLNa to total FLNa was nearly eight times higher in MCF-7/Cav1 cells, indicating that the heightened phosphorylation of FLNa cannot be explained exclusively by its increased levels. Ravid et al. concluded that caveolin-1 and IGF-I-induced phosphorylation of FLNa is mediated through the PI3K/Akt pathway, underscoring the complexity of FLNa's role in cellular processes and its potential impact on cancer biology [48].

### 4.5. Role of Filamin A in the Angiogenesis of Breast Cancer

Numerous studies have confirmed the involvement of FLNa in cell migration, observed in pathological and physiological contexts of blood vessel formation [5,57]. Zheng et al. demonstrated that hypoxia-inducible factor-1$\alpha$ (HIF-1$\alpha$) physically interacts with FLNa, promoting tumor growth and angiogenesis induction [58]. Under hypoxic conditions, increased FLNa fragmentation by calpain is noted, with the resulting C-terminal fragment interacting with HIF-1$\alpha$, facilitating its nuclear accumulation [58]. This cascade amplifies angiogenesis and tumor growth. Li et al. revealed that R5, a neutralizing antibody targeting roundabout guidance receptor 1 (Robo1), inhibits breast cancer progression and metastasis by suppressing angiogenesis through FLNa downregulation [59]. Their findings showed that R5 treatment hindered tumor growth in athymic nude mouse xenograft models with human breast cancer cells (MDA-MB-231) and angiogenesis in 4T1 breast cancer cells. Additionally, R5 treatment inhibited the migration of human umbilical vein endothelial cells (HUVEC) and impaired their ability to form tubular structures, concurrently reducing FLNa levels. Furthermore, overexpression of FLNa in the HUVEC cells counteracted the effects of R5 treatment [59]. Despite these promising results, it is important to acknowledge that this study only indirectly confirmed FLNa's role in breast cancer angiogenesis. The expression of FLNa in the vascular network around xenograft breast tumors in mice, treated with or without R5, was not compared, and the research on this aspect was solely conducted using the HUVEC cell model. Moreover, details regarding the functional association between Robo1 and FLNa remain unclear. Li et al. suggested a potential link between Robo1 and FLNa based on their ability to interact with actin [59]. Robo1 is a binding partner for the actin-nucleating complex, while FLNa is an actin cross-linking protein. Currently, there are no other reports on the engagement of FLNa in breast cancer angiogenesis.

This chapter aimed to present FLNa's impact on breast cancer development. All reported roles of FLNa in breast cancer are summarized in Table 1.

**Table 1.** Summary of FLNa roles in breast cancer.

| Role | Material | Conclusions | References |
|---|---|---|---|
| Expression | Breast cancer tissue | FLNa present in both normal tissue and non-malignant carcinoma | [39] |
| | | Exclusively cytoplasmic expression in all breast cancer samples<br>Primarily membranous with some cytoplasmic presence in normal tissues | |
| | | No significant relationship between FLNa levels and clinicopathological parameters | |
| | Breast cancer tissue | FLNa mostly undetectable in normal breast tissue and benign tumor samples | [40] |
| | | Increase in the percentage of FLNa-positive samples with cancer's progression | |
| | | FLNa mostly located in cytoplasm, especially at the cell edges, present in basal cells and the intercellular substance | |
| | | Correlation between FLNa levels and TNM stage, presence of metastases, vascular or neural invasion, and menstrual status | |
| | Breast cancer tissue | FLNa hardly noticeable in normal breast tissue and benign tumors | [41] |
| | | Primarily found in the cytoplasm of myoepithelial and basal cells, or between cells in breast cancer tissue | |
| | | Negative correlation between FLNa expression and tumor size<br>Correlation between FLNa and increased expression of progesterone receptors | |
| | Breast cancer tissue and MDA-MB-231 cells | FLNa protein expression in small number of samples | [42] |
| | | Primarily found in the cytoplasm, for MDA-MB-231 cells also at the cell membrane | |
| | MDA-MB-231 cells | Lower FLNa expression in cells transduced with a TET-off construct containing mir-200c | [43] |
| | MCF-7 cells | Increase of FLNa expression after knocking out mir-200c | |
| | MCF-7 cells | Lowering FILIP1L levels increases FLNa expression | [44] |
| Migration and invasion | MCF-10A and MDA-MB-231 cells | While not fatal, knocking out FLNa lowers cancer cells' ability to migrate | [46] |
| | Mice xenograft | Mice with FLNa knocked out showed no metastasis | |
| | MDA-MB-231 cells | Silencing FLNa decreases the rate of migration and invasion of breast cancer cells | [47] |
| | MDA-MB-231 and MDA-MB-436 cells | Inhibition of FLNa hinders cell migration and invasion | [42] |
| | Mice xenograft | Mice injected with FLNa-deficient breast cancer cells showed less frequent metastasis | |
| | MCF-7 cells | Expression of caveolin 1 mediated by its filamin A phosphorylation enhancing properties increased cells motility | [48] |
| | MDA-MB-231, MCF-7, BT-549, Hs 578T, MDA-MB-468, BT-474, ZR-75-1, and HMECs | FILIP1 mRNA expression shows negative correlation with invasiveness of the cells | [44] |

**Table 1.** *Cont.*

| Role | Material | Conclusions | References |
|------|----------|-------------|-----------|
| DNA repair and therapy outcome | MDA-MB-231, HCC38, Htb126 and HCC19337 cells | Knocking down FLNa increases sensitization to docetaxel but not doxorubicin | [55] |
| | Mouse xenograft | Mice with FLNa knocked down were more sensitive to docetaxel compared to control group | |
| | MDA-MB-231 cells | Elevated levels of FLNa increase sensitivity to paclitaxel | [56] |
| Regulation of gene expression | MCF-7 and MDA-MB-231 cells | siRNA knockdown of JUN in wild type MDA-MB-231 and MCF-7 cells carried out FLNa mRNA decrease | [43] |
| | MCF-7 cells | Increased levels of caveolin-1 increase FLNa expression Caveolin-1 and IGF-I-dependent phosphorylation of FLNa happens via the PI3K/Akt pathway | [48] |
| Filamin A as a breast cancer marker | Serum samples | Using markers with combinations of proteins that contained FLNa carried high sensitivity and specificity against breast cancer | [60] |
| | MCF-7, SK-BR-3, BT-474, BT-20, BT-549, and ZR-75-1 MCF-10A and HMECs | FLNa expression increased in media of breast cancer cells' media, while not found in normal HMECs or MCF-10A cells | [61] |
| | Breast cancer tissues | Lack of FLNa could be used as a prognostic marker for cancer metastasis | [42] |
| Angiogenesis | Mouse xenograft | Under hypoxic conditions calpain fragmentates FLNa, which interacts with HIF-1$\alpha$ facilitating its nuclear accumulation | [58] |
| | Mouse xenograft 4T1, HUVEC cells | R5 downregulates FLNa which in turn inhibits breast cancer progression and metastasis by suppressing angiogenesis | [59] |

### 4.6. Clinical Relevance of Filamin A in Breast Cancer and Future Perspectives

The clinical significance of FLNa can be considered in two distinct ways. Studies confirming the overexpression of FLNa in breast cancer cells compared to normal breast tissue suggest that it may serve as a breast cancer marker. Furthermore, because of its involvement in processes important for cancer progression, FLNa could also become a target in breast cancer therapy.

The significant correlation between FLNa overexpression and breast cancer has led to extensive research into its potential as a diagnostic marker for the disease. Fredolini et al. identified that among 56 proteins with elevated levels in the serum samples of patients with invasive ductal carcinoma (IDC), 32 were unique to IDC [60]. Remarkably, their study highlighted a panel of four proteins, including FLNa, which demonstrated a capability to distinguish breast cancer patients with 100% sensitivity and 85% specificity at early T1a stages [60]. The team further developed two models for breast cancer markers: Model 1, which included cofilin 1, alpha-2-HS-glycoprotein, and FLNa, achieved nearly 90% sensitivity and 80% specificity. Model 2, comprising Ras-related protein, integrin alpha-IIb, FLNa, and talin-1, showed 100% sensitivity and approximately 85% specificity [60].

In another study, Alper and colleagues utilized a novel anti-FLNa antibody, Alprer-p280, to detect FLNa secretion from various breast cancer cell lines, including MCF-7, SK-BR-3, and others [61]. On the other hand, FLNa was not found in the medium of normal human mammary epithelial cells, or the non-tumorigenic human mammary epithelial cells—MCF-10A [61]. This differentiation extended to tissue samples, where breast cancer tissues exhibited higher FLNa levels compared to normal tissues, with 85% of the latter showing negative results. Intracellular FLNa levels were significantly elevated in more atypical

breast cancer samples, with immunohistochemical staining revealing FLNa increases in the stroma of cancerous tissues and its presence in the lumina of vesicular compartments within carcinoma samples, a feature not observed in normal tissues. Moreover, FLNa levels were higher in the plasma of cancer patients, especially those with metastases compared to those with only primary tumors [61].

Jiang et al. explored the relationship between FLNa expression and clinical survival, focusing on the prognosis of breast cancer patients with distant metastasis [42]. Their findings indicated that patients negative for FLNa expression had a higher survival rate compared to those positive for FLNa. This led to the conclusion that the absence of FLNa could serve as a prognostic marker to assess the risk of breast cancer metastasis and that targeting FLNa expression might offer a therapeutic strategy to reduce tumor metastasis in affected patients [42].

The available literature reports discussed in this review highlight the pivotal involvement of FLNa in processes related to the progression of breast cancer, including cell migration, invasiveness, angiogenesis, and gene expression regulation, as well as DNA repair and response to therapy. This evidence provides a solid foundation for research into the use of FLNa as a therapeutic target. The comprehensive role of FLNa in these critical elements of cancer development makes it an attractive candidate for the development of targeted therapies that could offer more effective and specific treatment options for patients. However, it is important to note that further research focusing on various types of breast cancer is needed, as current reports suggest a varied impact of FLNa on therapy response for different types of cytostatics. Available reports indicate that FLNa downregulation enhances the effect of docetaxel, yet does not affect the response of cells to doxorubicin [55]. Conversely, a higher level of FLNa leads to an intensified response of MDA-MB-231 cells to paclitaxel [56]. This variability underscores the complexity of FLNa's role in breast cancer treatment and highlights the necessity for a more nuanced understanding of its function across different cancer subtypes and therapeutic agents. This insight clearly points to an excellent direction for future research, emphasizing the need to delve deeper into the differential roles and impacts of FLNa, potentially paving the way for more effective and personalized cancer therapies.

To our knowledge, there are currently no clinical trials that specifically focus on the role of FLNa as a therapeutic target in cancer patients. However, there has been a study investigating the use of FLNa in predicting early recurrence of hepatocellular carcinoma after hepatectomy. This retrospective study, conducted on samples from 113 patients, revealed a significant correlation between intra-tumoral and peritumoral immune reactivity to FLNa and early recurrence. This finding highlights the potential of FLNa not only as a biomarker for cancer progression, but also as a predictor of disease recurrence, suggesting an area for future research in the context of improving post-surgical outcomes for oncological patients [62].

The function of FLNa is also intricately linked to the protein's form as it may undergo cleavage in the hinge 1 and 2 domains by calpain, resulting in the release of a 90 kDa carboxyl-terminal fragment. This fragment plays a crucial role in mediating cell signaling and facilitating the transport of transcription factors into the cell nucleus. Calpeptin is a chemical inhibitor that targets both calpain 1 and 2. Studies conducted on cells, including of melanoma and prostate cancer cell lines, have demonstrated that the application of calpeptin leads to the inhibition of cell proliferation, migration, and colony formation [63]. The action of calpeptin was also preliminarily investigated on two breast cancer lines of different aggressiveness—MCF-7 and MDA-MB-231. The experiments showed that calpeptin causes a decrease in cell viability in both lines compared to untreated controls, which was associated with the induction of apoptosis and, in the case of MDA-MB-231 cells, cell cycle arrest in the S phase [64]. However, it has not been investigated whether the observed effect is linked to calpeptin's interaction with the cleavage of FLNa, nor how treating cells with the inhibitor affects other cellular processes such as migration. This presents an intriguing aspect worth exploring in future research.

## 5. Conclusions

Breast cancer research has uncovered the importance of FLNa in understanding cancer biology. This article explores the many roles of FLNa, from its well-known functions in the cytoplasm to its growing significance within the cell's nucleus. It provides a thorough overview of how FLNa influences the development, progression, and spread of breast cancer.

The FLNa protein plays a complex role in cancer cells. It can act as a scaffold, helping to bind actin in the cell's structures. It also regulates gene expression and DNA repair processes in the cell's nucleus. This versatility reflects the intricate ways FLNa influences cancer cell behavior. Furthermore, FLNa shows potential as a diagnostic and prognostic marker, especially for aggressive forms of breast cancer like triple-negative type. This suggests that FLNa could be an important breakthrough in the discovery of breast cancer biomarkers.

As we continue to unravel the mysteries surrounding FLNa in the context of cancer, it is clear that this protein is one of the key elements for understanding not just how breast cancer develops, but also the pathways to effective treatment and metastasis prevention. FLNa research highlights the importance of personalized treatment strategies, while insights gained from studies on FLNa offer promising ways for enhancing breast cancer diagnosis, prognosis, and treatment.

**Funding:** The present study was co-supported by research tasks within the framework of statutory activities (Nicolaus Copernicus University in Toruń, Faculty of Medicine, Collegium Medicum in Bydgoszcz).

**Conflicts of Interest:** The authors declare no conflicts of interest.

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
