# Peer review of "Role of Filamin A in Growth and Migration of Breast Cancer—Review"

_cimb, doi:10.3390/cimb46040214_

Round 1

Reviewer 1 Report

Comments and Suggestions for Authors

In their study, Patryk Zawadka and colleagues have provided insight into the multifaceted functions of Filamin A (FLNa), an actin-binding protein (ABP), and its implications in breast cancer. FLNa emerges as a pivotal player in cellular processes, exerting diverse effects that extend into the realm of breast cancer biology. While consensus exists among researchers regarding its involvement in promoting breast cancer progression and aggressiveness, variations in findings underscore the complexity of FLNa's role. Furthermore, elucidating the precise mechanisms by which FLNa influences critical aspects such as cell migration, invasion, and cancer progression remains an ongoing challenge, necessitating further investigation.

The study undertakes the task of evaluating FLNa's potential as a therapeutic target by comprehensively summarizing its pivotal roles in both normal cellular function and breast cancer pathogenesis. By delving into FLNa's intricate interactions and functions within cells, the research sheds light on its significance in driving malignancy and offers valuable insights into potential therapeutic interventions targeting FLNa. This comprehensive analysis not only enhances our understanding of FLNa's involvement in breast cancer but also underscores the importance of continued research efforts aimed at deciphering its mechanisms and therapeutic potential.

Certainly, here are some specific comments which will make this review as per the standard of this journal.

Specificity of Roles: When discussing FLNa's roles in breast cancer, it would be beneficial to specify the exact functions or pathways it is involved in. For example, how does FLNa primarily influence cell proliferation, migration, invasion, or another aspect of cancer progression?

Mechanistic Insights: The abstract mentions that the precise mechanisms through which FLNa affects cancer progression remain unclear. It would be helpful to briefly discuss some proposed mechanisms or pathways that have been suggested in the literature, even if they are not fully understood.

Functional Significance of FLNa: While the section effectively outlines FLNa's role in cytoskeletal organization and cell adhesion, further elaboration on how these functions contribute to cancer progression, metastasis, and therapeutic resistance would enhance the discussion.

Interactions with Integrins and Talin: The interaction between FLNa, integrins, and talin is well-described, but discussing the implications of these interactions for cancer cell migration, invasion, and metastasis would add depth to the analysis.

FLNa and Cell Proliferation: Although the role of FLNa in cell proliferation and mitosis is addressed, further elucidating how FLNa-mediated actin rearrangement and phosphorylation events influence cancer cell division and tumor growth would provide additional insights.

Regulation by Kinases: While the interaction between FLNa and PAK1 is discussed, expanding on how PAK1-mediated phosphorylation of FLNa contributes to cancer progression, particularly in terms of metastasis and invasion, would enrich the discussion.

Coverage of FLNa Interacting Partners: Acknowledging the abundance of FLNa interacting partners is important, but providing examples of specific proteins and their roles in cancer biology could enhance the reader's understanding of FLNa's diverse functions within the context of malignancy.

Other Cancer Types: Discussing the rationale for investigating FLNa in breast cancer specifically or its relevance in other cancer types would broaden the scope of the article.

Pharmacological Inhibitors and Genetic Modification: Exploring the use of pharmacological inhibitors or genetic modification to study FLNa's effects in breast cancer would provide valuable insights into its potential as a therapeutic target.

Clinical Relevance in Breast Cancer: Considering FLNa as a potential therapeutic target, briefly discussing any clinical implications or potential applications of targeting FLNa in breast cancer treatment would be informative.

Author Response

Dear Reviewer,

Thank you for your comprehensive review and constructive feedback. We appreciate the time and effort you invested in evaluating our manuscript, and we are grateful for the opportunity to enhance its quality and alignment with the journal's standards. Here are our responses to the comments:

  1. Specificity of Roles: We included a clearer description of roles and pathways involving FLNa in breast cancer progression and development.
  2. Mechanistic Insights: Even though the specific mechanism of FLNa involvement in breast cancer remains poorly described, we tried to provide a more detailed account of the roles and pathways that involve FLNa.
  3. Functional Significance of FLNa: We have elaborated on FLNa's impact on cytoskeletal organization and cell adhesion and how it contributes directly to aspects of cancer progression, such as metastasis and therapeutic resistance, providing a clearer link between FLNa's basic functions and its implications in malignancy.
  4. Interactions with Integrins and Talin: We expanded our description of FLNa's interactions with integrins and talin to explore their collective implications for cancer cell behavior.
  5. FLNa and Cell Proliferation: The section on FLNa's role in cell proliferation has been enhanced with further details on how actin rearrangement and phosphorylation events mediated by FLNa influence cancer cell division and tumor growth.
  6. Regulation by Kinases: We have provided additional information on how PAK1-mediated phosphorylation of FLNa contributes to cancer progression, with a focus on metastasis and invasion, enriching our discussion on the regulatory roles of kinases on FLNa.
  7. FLNa Interacting Partners: To better illustrate FLNa's diverse functions in malignancy, we included examples of specific proteins that interact with FLNa.
  8. Other Cancer Types: In the Introduction section we included a list of other types of cancers in which overexpression of FLNa was confirmed. However, as some great reviews are already available on the general involvement of FLNa in cancers, we decided to focus specifically on breast cancer, what we also emphasized at the end of the introduction section.
  9. Pharmacological Inhibitors and Genetic Modification: In a newly added “Clinical relevance of Filamin A in breast cancer and future perspectives” section we described a potential use of FLNa inhibitor – calpeptin in cancer treatment. In previous sections we also mentioned studies involving downregulation of the protein.
  10. Clinical Relevance in Breast Cancer: Finally, in a newly added “Clinical relevance of Filamin A in breast cancer and future perspectives” section we have discussed the clinical implications and potential applications of targeting FLNa in the treatment of breast cancer, underscoring its significance as a therapeutic target.

We believe these revisions address your concerns and significantly enhance the manuscript's contribution to the field. 

Sincerely,

Maciej Gagat

Reviewer 2 Report

Comments and Suggestions for Authors

In the current review, the authors have discussed the roles of Filamin A in both normal and cancerous cells with a focus on breast cancer. While the review is well presented I have some comments that could improve the overall quality of the manuscript/

1. In the abstract, the authors mentioned that FLNa is a protein that plays various roles in cells and consequently in breast cancer. I think it is not a role that a protein with important cellular roles would also involved with cancer. Could the authors comment on that?

2. Filamin A has been studied under different types of human cancers and not only breast cancer. Authors should mention that briefly by the end of the introduction section then they can show that the focus of their review would be on breast cancer.

3. The authors have not mentioned the possibility of selecting Filamin A as a target for antitumor therapy. Are their previous trials or studies that used Filamin A inhibitors as anticancer agents?

4. By the end of their review, the authors should mention the future prospects of studying Filamin A in the field of oncology.

Comments on the Quality of English Language

Moderate editing of English language required

Author Response

Dear Reviewer,

Thank you for your insightful comments and suggestions regarding our review article. We appreciate the opportunity to address the points you've raised to enhance the manuscript's quality.

  1. Thank you for this valuable comment. Indeed, the mentioned fragment of the abstract was ambiguous and required further clarification. We have since rephrased it to improve the clarity.

“Filamin A (FLNa), an actin-binding protein (ABP), is involved in various cellular processes, in-cluding cell migration, adhesion, proliferation, and DNA repair. Overexpression of the protein was confirmed in samples from patients with numerous oncological diseases such as prostate, lung, gastric, colorectal, and pancreatic cancer, as well as breast cancer.”

  1. We strongly agree that this information should be included in the text. Accordingly, following your recommendation, we have incorporated it into the introduction.

“The involvement of FLNa in cancer development and progression was confirmed in numerous types of oncological diseases including prostate, pancreatic, lung, colorectal, gastric as well as breast cancer compared to normal tissues. However, an interesting aspect of the protein's function is also its localization within the cell, which likely has significant implications for the cancer progression. Studies suggest that high cytoplasmic expression of the protein favors invasive-ness, migration, and adhesion of cells, whereas nuclear expression affects gene expression and leads to a reduction in the cells' migratory abilities”

  1. Thank you for this comment. In a newly added “Clinical relevance of filamnin A in breast cancer and future perspectives” section we incorporated available information on clinical trials in oncological patients involving FLNa.

“To our knowledge, there are currently no clinical trials that specifically focus on the role of FLNa as a therapeutic target in cancer patients. However, there has been a study investigating the use of FLNa in predicting early recurrence of hepatocellular carcinoma after hepatectomy. This retrospective study, conducted on samples from 113 patients, revealed a significant correlation between intra-tumoral and peritumoral immune reactivity to FLNa and early recurrence. This finding highlights the potential of FLNa not only as a biomarker for cancer progression but also as a predictor of disease recurrence, suggesting an area for future research in the context of improving post-surgical outcomes for oncological patients”

  1. Thank you for this valuable suggestion. To address it, we added “Clinical relevance of filamnin A in breast cancer and future perspectives” section, which describes possibilities for using FLNa as a breast cancer marker or therapeutic target, as well as future perspectives for studies.

Moreover, the entire manuscript was carefully evaluated and reviewed by a certified English speaker.

We are grateful for your constructive feedback, which we believe will significantly improve the quality of our review.

Round 2

Reviewer 2 Report

Comments and Suggestions for Authors

The manuscript can be accepted in the current form

Comments on the Quality of English Language

Minor editing of English language required